# One4all—A New SCADA Approach

**DOI:** 10.3390/s22062415

**Published:** 2022-03-21

**Authors:** Bogdan Vaduva, Ionut-Flaviu Pop, Honoriu Valean

**Affiliations:** 1SCADA/GIS Department, S.C. VITAL S.A., 430011 Baia Mare, Romania; ionut.pop@vitalmm.ro; 2Automation Department, Technical University of Cluj-Napoca, 400114 Cluj-Napoca, Romania

**Keywords:** SCADA, supervision, sensing system, acquisition, multi view project, one4all

## Abstract

The main purpose of this paper is to introduce a new concept, named “one4all” in the realm of SCADA (Supervisory Control and Data Acquisition) systems, used by a regional company (particularly a water supplying company) for managing the different views of its users. As a secondary purpose, the paper presents an integration of such an SCADA system with a GIS (Geographical Information System) system. All the regional water supply companies in Romania manage water and wastewater networks, many sensors and actuators, dozens of water pump plants, several water treatment and wastewater plants, tanks and many hydrophores in different parts of their operating range. Due to the wide geographical operating range, an SCADA system needs to be put in place, but the management of such a system in a traditional way is hard to implement, especially when the human resource is low. The methodology presented in this paper, involving adding helper tables and dynamic template windows within an SCADA tool (“one4all” tool), will show how efficiently the human resource can be used. Additionally, the paper shows that companies as described above, can use a single SCADA system that generates different views for all the managed sub regions instead of different systems for every sub region. Implementing only one SCADA system built with the concept “one4all” in mind, and integrating it with a GIS system that is built on the same principle, represents a new approach that will bring value to any organization willing to adopt it. The concept of “one4all”, implemented as a software tool for an SCADA system, is a new concept that will help any developer to easily build applications that generate different views for different users based on their permissions and their operating range. Finally, the paper presents some examples of the same concept, implemented in a different vertical (GIS) and programming language, thus presenting that a “one4all” concept is viable and helpful, bringing value to the information technology industry.

## 1. Introduction

Romanian water supply and wastewater companies are organized as regional operators serving a geographic area of one or more counties. Having such a wide operating range from the SCADA [1,2] point of view, multiple local views, either HMIs—Human Machine Interfaces—or local SCADA projects, must be displayed, usually being integrated in a centralized control room. That room should be easily accessible by the management staff, allowing them to take informed decisions. Furthermore, having such a control room can allow operators to inform the specialized teams about items that are in error and what kind of errors they will deal with.

The paper is focused on the SCADA system of the regional operator for the county of Maramureş and shows what has been done there and how that work could help others. SCADA offers the possibility of receiving information from field equipment and sending back a limited set of instructions. SCADA is a bi-directional system that allows not only monitoring but also taking action by sending commands to an SCADA device.

SCADA is a worldwide spread concept that is implemented in a wide range of industrial processes. An SCADA system is a mix of software and hardware elements that allows industrial organizations to control their processes locally or remotely [3].

All SCADA systems are crucial for industrial organizations, either public or private, because they help to maintain efficiency. Furthermore, an SCADA system processes historical data for smarter decisions, and communicates system issues to the intended audience to help mitigate any downtime [3].

A basic SCADA architecture involves programmable logic controllers (PLCs) and/or remote terminal units (RTUs). PLCs and RTUs are microcomputers that communicate with an array of objects such as factory machines, HMIs, sensors, and end devices. SCADA software processes, distributes, and displays the data, helping operators and other employees to analyze the current or historical data and make appropriate decisions. SCADA systems are the backbone of many modern industries, such as: energy, food and beverage, manufacturing, oil and gas, power, recycling, transportation, water and waste water, and many more [4].

The regional operator began building its SCADA system by creating an SCADA project that integrated (at the beginning) just a few elements such as: two big wastewater treatment stations (Baia Mare wastewater treatment station and Sighetu Marmatiei wastewater treatment station), one water treatment station (Baia Mare), a few tanks and many hydrophores and pumping stations. Then, the regional operator extended this activity in other parts of its operating range, so a new set of elements needed to be integrated, but this expansion was set to happen on a regular basis for the next period of time. That widening of its activity involved more people that daily had to view/follow the SCADA system, but those people did not need to have access/view elements from other regions of the company’s operating range [5].

On the other hand, the GIS department had a web-based application that showed different views for different users using a concept named (by the authors) “one4all”. The “one4all” concept states that a basic framework, smart enough to generate the needed output, can be used to build an entire software application, without using other software elements. Building an application in such a fashion, where all the views are generated (not coded) through the use of the building block, will allow different users to have different views. The authors asked themselves if that concept can be extended in the realm of SCADA; thus, having a tool (building block) that generates different views for different users based on their permission into the system [6,7]. 

In the SCADA realm, the “one4all” concept means that we have one tool that unifies a set of templates that can be used to generate the views of the entire SCADA project. This outcome means that SCADA developers will no longer need to create a screen because they have a tool that generates that screen based on their description. Thus, an SCADA project that uses a “one4all” style will be able to generate different views to different users based on their permission into the system. From the “one4all” point of view, all (most of them will be enough) the SCADA windows must not be hard coded by the developers but generated by its “one4all” tool/framework.

For the study company, the “one4all” concept migrated from the programming realm. The GIS department had an application built on a “one4all” framework. That framework used helper tables for generating its views. 

Furthermore, we will try to introduce the readers to the new concept. Nowadays, programmers write source code for inserting, editing and deleting records of a relational table. The majority of commercial relational databases include a specific management tool that offers such possibilities, and most database programmers take this ability as granted. When it comes to real life applications, programmers use an Object Oriented (OO) paradigm to build user friendly windows/screens/forms for database operations. The “one4all” concept allows users to generate application views by using a “low-code” approach by simply setting the name of the desired table/view (to be displayed). Those views will further be used for building application windows. Having a framework that allows building CRUD (Create Read Update Delete) views will allow both programmers and users to rapidly modify an application. Furthermore, if the “one4all” framework can be integrated with user permissions and roles, more flexibility is gained [8].

The mentioned framework started an idea of building such a tool in the real SCADA. The future tool will need some kind of configuration repository, so a similar set of helper tables must be used for the SCADA one4all concept also. The GIS framework is written as a low-code framework, but such an approach is not applicable for SCADA because of the way an SCADA system is built [8].

The current developments in the GIS field make the idea of linking the elements from the SCADA views to the actual geographic position of those elements interesting. The paper presents such a system, which integrates an existing SCADA with the GIS and links the elements of the SCADA (plants, sensors) to their geographical position by providing the user access with respect to the one4all principle.

## 2. Migration of Traditional SCADA Systems to “One4all” Ones

### 2.1. Selecting a SCADA Software

At the beginning, our company did not have an SCADA department and the management assigned to the GIS department the task of implementing the future SCADA system. At that time, the GIS department was already using a web-based application with a “one4all” style framework. Further to this, we looked for an SCADA software that could allow us, in the future, to build a system in a similar manner, but making such a choice was hard. Another important aspect was the ability to integrate an unlimited amount of data tags. 

The company ended up choosing Ignition from Inductive Automation [9], which is a supplier of web-based industrial automation software based in Folsom, CA, USA. Ignition is a server software that acts as the hub for all the data tags of a company. 

We also looked at other companies such as Siemens and their product, called WinCC [10], but we gave up very quickly because of licensing costs. The licensing costs of any SCADA software also include the licensing costs of the operating system and the number of data tags that can be added into the system. We wanted to keep those costs at a minimal value, so we added a few restrictions for the SCADA software and one was for it to be Linux-based. Ignition’s Inductive Automation platform allows us to build an SCADA software, independent of the operating system, with an unlimited amount of data tags.

### 2.2. The First Attempt

The company’s first attempt to build an SCADA system happened in 2016 and was conducted by an external company called ADASA (https://www.adasasystems.com, accessed on 15 February 2022), through their local branch. ADASA gave the company a small project that integrated just a few SCADA elements. Furthermore, that SCADA project was not integrated in any way with the GIS data and was built in a traditional fashion by the extensive use of window templates. A template windows in an SCADA system has all its elements in place, which means it is not dynamic. Every time a template is used, changes to the window that uses the template are made.

The architecture of that SCADA system was as follows:Lower Level—which contains water pumping stations (fresh and waste), reservoirs, monitoring sensors (water pressure, chlorine, water level for tanks), all these being spread on the operation area of the company.Middle Level (local dispatch)—consisting of SCADA stations at the water treatment plants and water sewage plants.Higher Level (regional dispatch)—representing the decision-making level of the SCADA Regional System. It is responsible for implementing the communication system in the field and local dispatch units, collecting all the data, processing it in real time, managing databases, alarms and archiving the information.

Another issue faced by our study company was the lack of specialized personnel in the SCADA department. At that time, the company had only one person who could work on the SCADA system, but that person was already caught up in some other projects. After a while, the company successfully found one person to work with the Ignition software and, as a result, the new SCADA department had one actual developer. Having only one actual developer and a huge number of elements to be integrated into the SCADA system proved to be a challenge for the company.

Developing an SCADA project in a traditional fashion was clearly not a solution for the study company because by the time an SCADA element was integrated into the project, at least two more were standing in the queue to be added. The use of window templating was to help, but for a long run, templating could became a hassle. The use of a “one4all” framework stood out as a way for the future.

## 3. The “One4all” Concept

In this chapter, we aim to show how the concept of “one4all” was implemented within the SCADA realm and then how an SCADA project could easily be connected with the GIS system, due to the use of the “one4all” concept. 

The use of templating was not a solution for our study company because of the number of elements to be included into the SCADA system and the number of developers involved. SCADA templates are better if the elements within can be dynamically added. 

The concept of “one4all” from the study company’s SCADA point of view, means having dynamic templates gathered in a tool, able to generate the desired views. Having the views generated rather than being built will help developers build faster, better and easier. Furthermore, different SCADA users will have access to different views.

After the completion of that first project (roughly one year), the company’s SCADA department sat down and reassessed the future of the SCADA system and came up with a few requirements, such as:The SCADA system should be able to export some of its data tags through custom web services [11].The SCADA system users should have different views depending on their qualifications and position.The SCADA system users should be integrated with GIS data and users.The SCADA system should be able to run on mobile devices as well.The SCADA system would be easy to scale up in the future.

The next step was the rewrite of the project from the ground up, to be a complete functional solution for the water supply company’s management. The used data transmission protocols were set up to be OPC-UA [12,13,14,15] and Modbus TCP/IP [16]. The company’s SCADA solution is a web thin client and the project has to be downloaded from a gateway for every client (user) in order to be accessed. The newest Ignition version is a web-based solution, which means that a future migration to that new version will allow SCADA developers to be more creative.

We mentioned that we chose Ignition to be the SCADA software, but Ignition had different repository types for its historical data (database servers’ types) and from those database server types, the company choose PostgreSQL [17]. The PostgreSQL was chosen because it was the same database server used by the company’s GIS department and it was thought that it would ease any future GIS integration.

The next step was to set up a plan for the new SCADA system, which involved the need of a “one4all” SCADA tool. 

The new “one4all” tool was envisioned as having the following structure (Figure 1):

The structure above shows the need for a new set of tables as somewhere to keep the description of the future generated views. 

## 4. Implementation

The “one4all” tool was thought to integrate a set of template windows and have as a parameter only the name of the SCADA element that needed to be shown/generated in the SCADA’s client window. 

With that in mind, a new schema was added into Ignition’s database server. Into that newly created schema the following tables were added: “tbl_user”, “tbl_role”, “tbl_window”, “tbl_station”, “tbl_station_type”, “tbl_location”, “tbl_menu”, “tbl_report”, “ “tbl_permission_role_location”, “tbl_permission_role_window”, “tbl_permission_user_location”, “tbl_permission_user_window”, “tbl_permission_user_menu”, “tbl_permission_user_report”, “tbl_permission_user_station_type” and a few other helper tables that we will not mention here because those were mainly used for custom purposes. The above tables were a modified and stripped version of the helper tables used by the GIS “one4all” framework, and by doing so, the authors started the migration of the “one4all” concept from the programming realm to the SCADA’s reality. 

The new proposed approach was to use Ignition as a programming tool and not only as a design tool.

Next, we will describe each table’s purpose:“tbl_user”—will hold the users for the SCADA application and will be in sync or linked to Ignition’s gateway users and GIS users.“tbl_role”—will hold SCADA application roles. We defined only two roles.“tbl_window”—will hold the names and properties of SCADA application windows. Usually, these windows will display a process.“tbl_station”—will hold the names and properties of stations/elements displayed in the SCADA application windows. For the rest of our paper, we will refer to these stations/elements as stations.“tbl_station_type”—will hold the station type. Each station has a type and for our water company we tried to keep the number of station types to as few as possible because having a small number of station types will mean easier maintenance. Furthermore, each station type has attached a predefined set of data tags, allowing SCADA developer(s) to build some kind of templates.“tbl_location”—will hold the name and geographical reference of the station’s location.“tbl_menu”—will hold the elements of the menu that will be displayed in the application.“tbl_report”—will hold the names of the application’s reports.“tbl_permission_*”—are the tables for setting the user’s permissions throughout the application.

The addition of our helper tables to Ignition’s schema gave us a few new possibilities, one to implement a “one4all” SCADA tool and one to easily link any SCADA element to the GIS system. One other possibility was to have different views for different users, based on their permission and role. We will not go into the details of the helper tables because they represent a way of building a “one4all” SCADA tool and not an industry standard.

The new SCADA system was envisioned as having two base elements (as well as the elements that Ignition software was already providing):The “one4all” tool/framework that incorporates all the dynamic templates used by the SCADA system.The database server that contains the description and permission of generated SCADA system screens/elements.

Further to this, we want to show the steps taken by the authors to build the “one4all” tool.

After setting up the above plan, the authors moved forward by creating the specified tables in the PostgreSQL database server, thus creating the premises for the new SCADA system to be built using a “one4all” tool. The next step of our methodology was to identify station type templates for our displayed elements. This step involved identifying what elements should be displayed for each station type from a SCADA point of view. In this process of identifying station types, we found that we had, throughout the company, a set of elements that did not fit into any station type. Those elements were sensors (pressure and tank level sensors) that the company was using for monitoring the status of its water network. For these sensors, we used OSH (Open Sensor Hub) [18], for which we built a custom driver that allowed us to specify where to save the historical data. The architecture for gathering the historical data was as follows:Each sensor was connected to a serial to Ethernet converter device.We set up an OSH server and installed our custom driver.We made the appropriate changes to the custom elements (sensors) added into the OSH server. Those changes involved setting the SCADA database as the repository for our historical data.The final step was to integrate those sensors into the SCADA software.

Before going further, we want to say a few words about the sensor types used by our company. The company uses the following main sensor types: pressure sensors, temperature sensors, chlorine sensors, PH sensors, level sensors, status (open/close) sensors, and flowmeters. We also tried to standardize each sensor type, which means for each sensor type we chose one supplier that we thought would fit our needs and obtained the sensors from them.

Next we consider the pressure sensors (Figure 2a,b) and discuss them. For these types of sensors, we chose the following suppliers: NIVELCO, MEINSBERG, HUBA CONTROL, and WIKA. Each pressure sensor converts the physical quantity “pressure” into an industry-standard signal (electrical current, usually 4–20 mA). That electrical current is read by a MOXA [19] box and exposed through the Modbus TCP/IP standard into the SCADA network.

If we consider another sensor type and discuss it. For this discussion, we will use level sensors. Our study company uses two kinds of level sensors: a 2-wire loop-powered ultrasonic level measurement transmitter for measuring storage vessels from SIEMENS and a TL-136 liquid level transmitter water level sensor detector 12–32VDC 4–20 mA signal output (Figure 3). 

Once we carried out the above steps, we were able to display those sensors in the Ignition software in an easier manner. After a while, the company “discovered” the MOXA devices and reassessed the way those sensors were integrated into the SCADA. MOXA devices are serial media converters that allow devices with different serial interfaces to communicate effortlessly. 

Once the problem of the custom sensors was solved, the authors concentrated on the station types. Each station type was using a different PLC (programmable logic controller) type and in this step, the authors tried to conduct a standardization of the used PLC types. Each PLC type had a set of data tags being used. On this step, we want to show an example of standardizing the data tags; standardization that would allow in the future a PLC to be changed without changing the SCADA project. 

In Table 1 we show an example of data tags for a PLC type.

For each station type, the authors built a dynamic template that had its elements dynamically turned on/off depending on a set of parameters. All the dynamic templates were further gathered in a tool that helped with developing the SCADA system. 

As result, integrating a station was as easy as follows: it must be declared in the gateway page with its IP address and the required data tags (which can be copied from another station of the same type),then it must be registered in the “tbl_station” table with all the required properties. At the same time as the station is registered into the “tbl_station”, a new Ignition folder is created within the gateway. Furthermore, the data tags are taken from a template.Finally, the new station windows will be generated into the SCADA client using the internal templates of the “one4all” tool.

We stated in our paper that we wanted different views for different users, but what does this mean? Our water supply company has a wide geographic operating range, which means it has stations all over the region, but one or more stations were managed by at least one person who was different from the central dispatcher of the company. Moreover, each sub region has a manager that manages that sub region. By sub region we usually refer to a city/town. Each manager should be able to see only its region’s stations/elements but not the others (Figure 4 and Figure 5). Here, the concept of “one4all” could give us the output needed by our SCADA software. A “one4all” implementation will generate a station window by simply setting the name of that station into the tool and the result will be different depending on the user. Moreover, the same “one4all” tool was extended to integrate other window types and not only the station’s windows.

The “One4all” concept was thought to allow different users to have different views. In order to achieve this, we linked each user to a specific view/window, and if the user did not have a view set up, a default view/window would be displayed. Doing so allowed the company to have different views for different users; but what about the reports? To have the reports linked to a user/role, the custom tables “tbl_permission_*” were used. 

One other question that can be raised here is related to the fact that user management could became a hassle. To answer such a question, the user management windows had a feature called “Duplication”, which allowed the SCADA manager to duplicate a user or an SCADA element. How does an SCADA element work? If a new station needs to be put in place, the SCADA manager goes into the station managing window and selects a station to duplicate and then does it. By doing so, the new station is created with all the properties of its parent and is available for the SCADA client at the same time. 

Another goal stated in our paper was related to the fact that the SCADA project should be able to run on mobile devices; but what happens if a station window is open on a mobile device? The answer is that the station window will either be very small or the display is unreadable. To handle this new request, the “one4all” tool’s template windows were set up with two views depending on where that window would be displayed (computer or mobile device (Figure 6).

The default reading interval for SCADA’s historical data is 1 s, but for a better storage, management was set to 10 s. In order to achieve a stable SCADA system, the functions and methods from the backend were distributed between Ignition’s scripts and the PostgreSQL stored procedures/functions. The application obtains its data from the PostgreSQL database, asynchronously, in order to populate the graphical templates and, thus, not slowing down the user interface.

The presented SCADA solution was later integrated with the web-based GIS application allowing the company to easily pinpoint an SCADA element. For example, the water network map within the GIS application contains the water pumping stations’ geographical position. The GIS application can extract for each pumping station the total invoiced water over a period of time, but from SCADA, the pumped volume of water for the same period can be extracted, creating a comparison, similar to the one in Figure 7.

Regarding the user’s security, several layers were put in place. That means, if a new user is created in the gateway administration page, that user cannot log into the SCADA application if the Administrator did not grant him access from the SCADA security module.

As well as the user’s security, the presented SCADA system is now on a separate VLAN with no Internet allowed. Of course, there are some entrance points within that VLAN, but those are monitored by the company’s IT team, which makes the SCADA system as secure as it can be. The company also evaluated proprietary SCADA security software but the costs were too high, so those were postponed for the time being.

## 5. Results

The “one4all” SCADA tool was created and the “one4all” term could be attributed to the tool because it was the only tool used to build the entire SCADA project. The majority of SCADA windows were generated by the “one4all” tool, allowing different views for different users (Figure 4, Figure 5 and Figure 6). In the following figure (Figure 8), an example of a pumping station’s dynamic template is presented. Beneath the dynamic template is an example for the regional operator in the county of Maramureş. Within the dynamic template there is a property that allows developers to specify the name of the pumping station and the view is generated (elements turned on and off).

The “one4all” tool allowed the regional operator to rewrite the entire project within the timeline of 3 months, even though the regional operator was expanding its activity, which meant adding new elements to the SCADA project. The expansion of the regional operator involved also adding new template windows to the “one4all” tool, based on new requirements. It must be shown that even though the paper presents the results for the regional operator in the county of Maramureş, the “one4all” concept can easily be translated to other industries and even to other verticals such as programming.

The same concept of “one4all” in the programming realm (JavaScript/Angular 2+), has the following usage:

<homerp-dynamic-form 

   [tableName] = “‘tbluser’”

   [insertCollapsed] = “true”

   (insert) = “insert($event);”

   (edit) = “edit($event);”> </homerp-dynamic-form>, and two of the resulting views are in Figure 9 and Figure 10.

This company’s SCADA project was initially intended to be used in read-only mode, which means no PLC commands were implemented. After a while, the necessity to send commands to different PLCs arose and was added. Sending commands to PLCs can be easily implemented using the Modbus TCP/IP standard, thus making it just a click away from any user. Of course, all the commands have to have a higher security level, which means that a command can only be sent after user retype their own password.

Talking about the performance of the SCADA project built using the “one4all” tool, we must mention the total number of data tags that are in use. That number was, at the time this article was written, around 27,000 data tags and the application’s loading time for a computer view was about 7 s and that for a mobile device was about 10 s, but those numbers differ based on the computer’s and mobile device’s hardware specifications.

## 6. Conclusions/Future Works

The SCADA project was built using a “one4all” tool, so it can be easily implemented to other companies such as electrical, gas or any type of production. This will mean that we can use the entire SCADA project in a different vertical by adding new template windows into the “one4all” tool or use it as a module in a different SCADA project (user administration module, report module). The only current restrictions are related to the fact that the presented “one4all” tool was implemented onto the Ignition platform. One improvement that could be performed here is to switch to Ignition’s custom objects. Doing so means to obtain the appropriate licensing rights from Inductive Automation.

Building projects with a “one4all” concept in mind will allow any SCADA developer or programmer to build quickly and efficiently without quality loss. Moreover, using the new Ignition version, which is pure web-based, will allow developers to merge the presented “one4all” tool with other similar frameworks. 

Throughout our paper we have shown that the concept of “one4all” can be applied in different verticals (water and wastewater management, programming), and the use of it brings added value to any company willing to try it. Furthermore, the presented results, which are specific for a regional operator in Romania, can be extrapolated to other companies. 

Regarding scalability, the authors think that the presented SCADA system is a scalable one because even ordinary SCADA users can add SCADA elements/screens within the SCADA system. That is possible because SCADA elements/screens are generated and not built by SCADA developers. Furthermore, the system can be easily updated with new models (within the “one4all” tool), models that are instantly available to ordinary users.

One other note that can be made here is about the sensor network. At the current time, the “one4all” framework is used to display the sensor network of the water company in the Maramureş county, but it has not received any business intelligence, mostly because the company did not have resources to go in that direction. Adding business intelligence to a sensor network or other elements could be something to be carried out in the future.

As a final conclusion, the main contribution of this paper is the presentation of a concept named “one4all” (derived from dynamic templates), used to generate SCADA views to different users based on their permissions and roles.

## Figures and Tables

**Figure 1 sensors-22-02415-f001:**
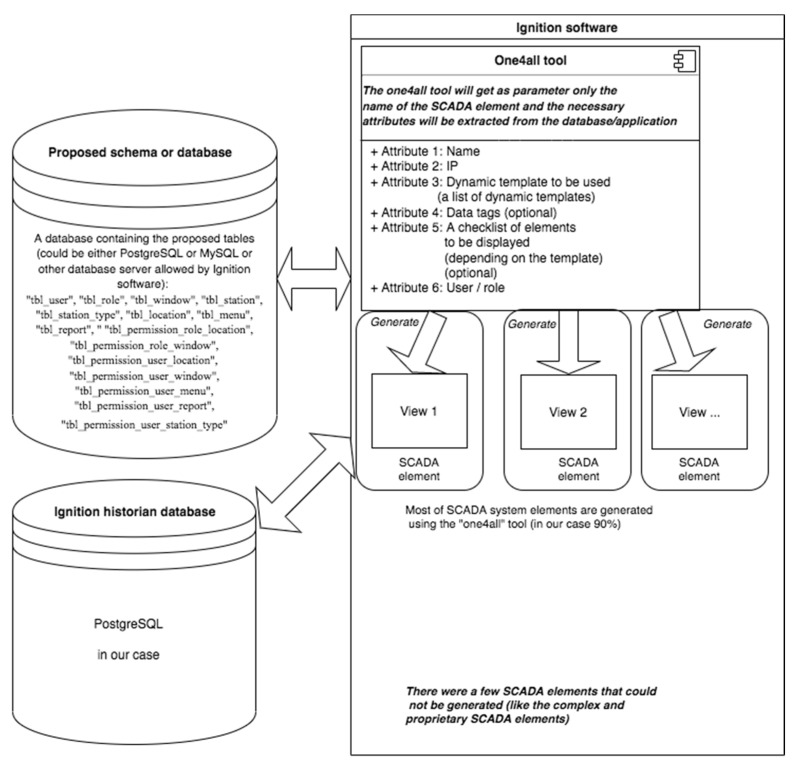
Schema of the “one4all” tool.

**Figure 2 sensors-22-02415-f002:**
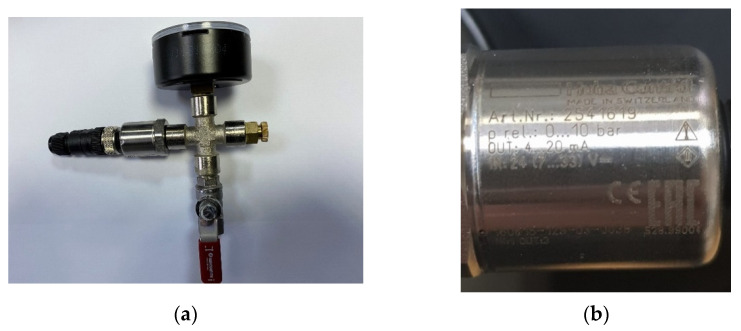
Huba Control sensor type: (**a**) overview image of the used sensor ensemble; (**b**) detailed view of our sensor.

**Figure 3 sensors-22-02415-f003:**
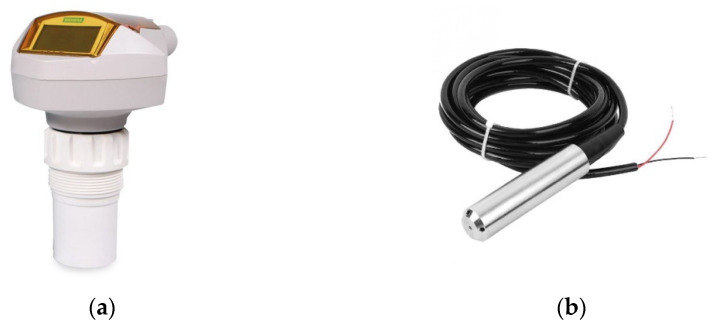
Level sensors: (**a**) Siemens ultrasonic level sensor; (**b**) TL-136 liquid level transmitter.

**Figure 4 sensors-22-02415-f004:**
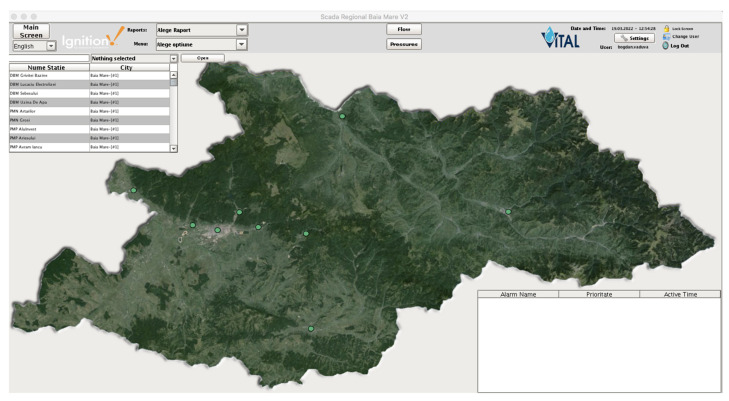
Default view for a user with Administrator role (green dots represents the status of the current operated locations/cities).

**Figure 5 sensors-22-02415-f005:**
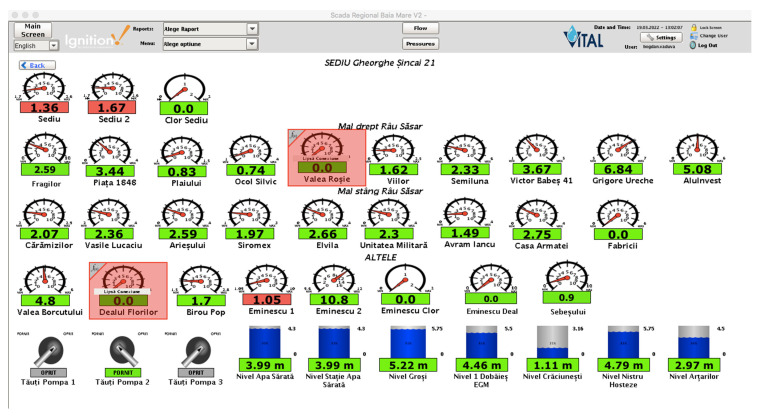
Custom view for a user with Operator role. (green means that the value shown it’s within normal limits, red means otherwise).

**Figure 6 sensors-22-02415-f006:**
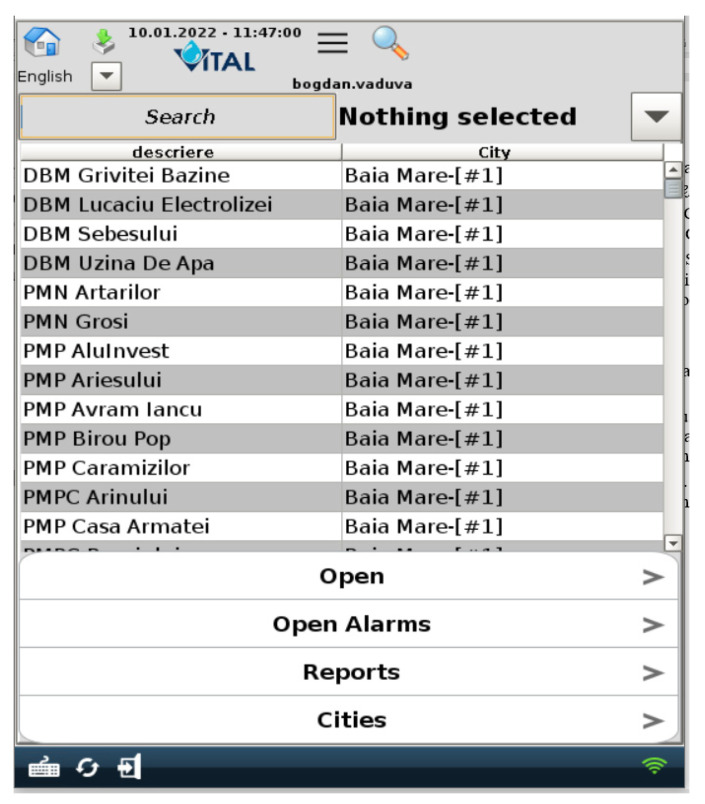
Default mobile view for both Administrator and Operator role.

**Figure 7 sensors-22-02415-f007:**
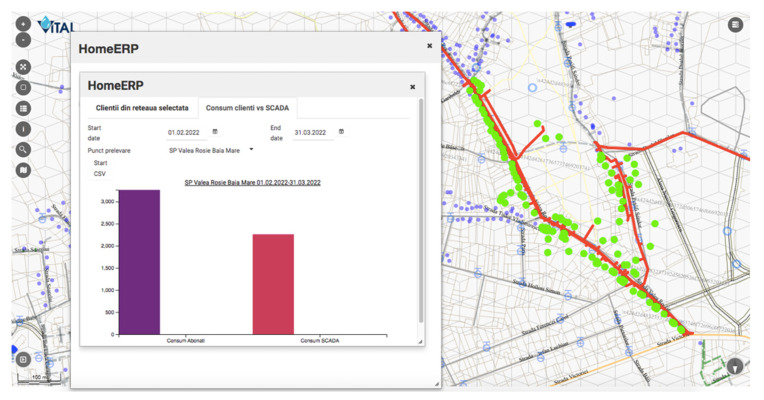
Comparison between invoiced and actual consumption. (The highlighted pipe network has the red color; the clients are the green dots).

**Figure 8 sensors-22-02415-f008:**
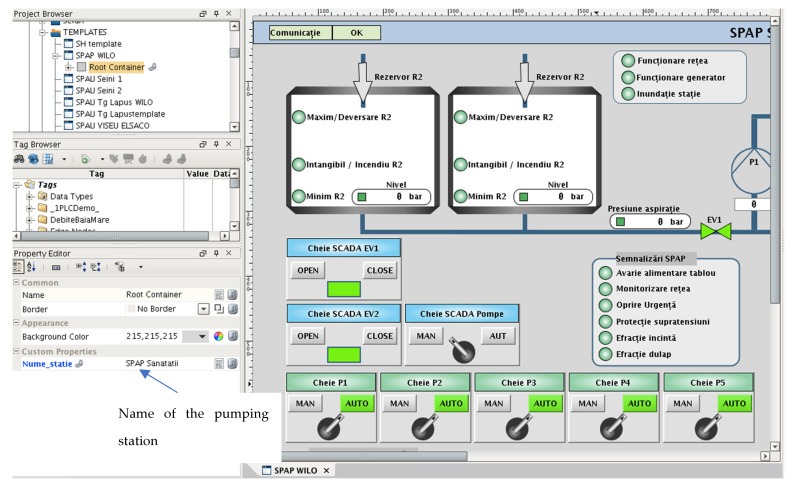
Dynamic template window for pumping stations.

**Figure 9 sensors-22-02415-f009:**
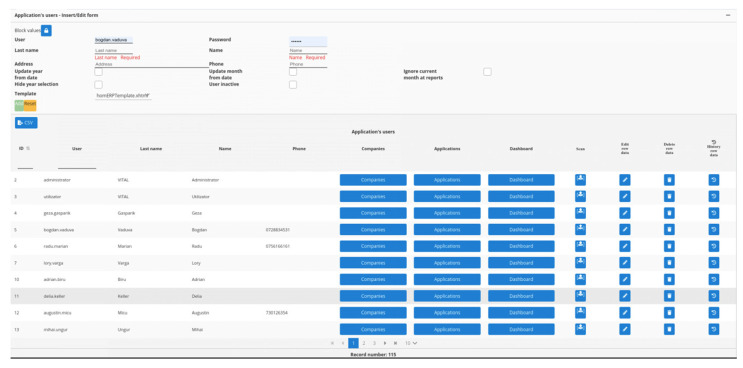
JavaScript/Angular 2+ “one4all” framework example—default view.

**Figure 10 sensors-22-02415-f010:**
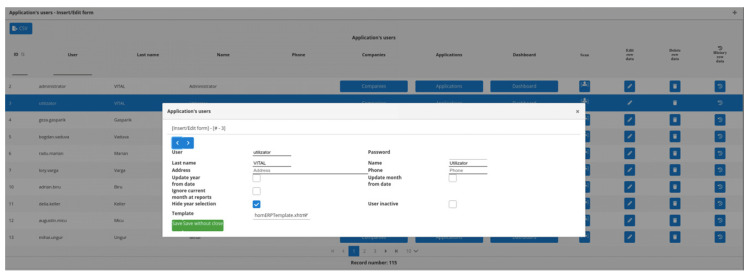
JavaScript/Angular 2+ “one4all” framework example—edit view.

**Table 1 sensors-22-02415-t001:** Data tag standardization. (An example for a pumping station).

Data Tag Addresses—SPAU				
*Station Parameters–Applicable Function Code: 03H–Read Holding Registers, 06H-Write Single Register; 10H-….*				
Address	BIT	Description	Access Type	Data Type
0		SCADA Communication	Read	UINT16
1		Level	Read	UINT16
2		Pump work level (0–100%)	Read	UINT16
3		Pump no (2 pumps; 0 = P1/P2, 1 = P1 + P2)	Read	UINT16
4		Minimum value for converter frequency	Read	UINT16
5		Maximum value for converter frequency	Read	UINT16
6		Pump working hours	Read	UINT16
7		Signal for shutting the station off/on	Read/Write	UINT16
8		Minimum level to stop the pumps	Read/Write	UINT16
9		Maximum level to start the pumps	Read/Write	UINT16

## Data Availability

Not applicable.

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
