# Peer review of "One4all—A New SCADA Approach"

_sensors, 2022, doi:10.3390/s22062415_

Round 1
Reviewer 1 Report
The paper present a framework to implement a monitoring plateform of SCADA systems. The framework is deployed for water management.
Overall, this article is well written and presents a lot of details about the proposed system. I would like to see some supplementary illustrations, such as a schema of the system architecture or a schema of the data base model.
My main concern for this article is the type of work presented here.
It does not look like a research article but more like a technical report on one particular system.
Concerning the literature review. The positionning of one4all framework seems conceptually near to business intelligence. This type of systems could be considered to manage the sensor network ?
Concerning the framework itself, is the code available somewhere ? Due to the very technical point of view of this article, I would prefer to have access to the framework, to be able to reproduce the proposed plateform.
Author Response
We appreciate you and the reviewers for your precious time in reviewing our paper and providing valuable comments. Your valuable and insightful comments, led us to improve our paper. The authors have carefully considered the comments and tried our best to address every one of them. We hope that the manuscript, after careful revisions, will meet your high standards. The authors welcome further constructive comments if any.
Below we provide the point-by-point responses. All modifications in the manuscript have been highlighted in red.
Thank you again for your time and suggestions.
Response to Reviewer
Comments and Suggestions for Authors
The paper presents a framework to implement a monitoring platform of SCADA systems. The framework is deployed for water management.
Overall, this article is well written and presents a lot of details about the proposed system. I would like to see some supplementary illustrations, such as a schema of the system architecture or a schema of the data base model.
Response: It was our pleasure to have you as our reviewer, due to your helpful and insightful comments. We’ll try to add to the manuscript the required information, so it can meet the standards required by any scientific paper.
The system architecture is as follows:
- One “one4all” tool/framework that incorporates all the dynamic templates used by the SCADA system
- One database server that contains the description and permission of generated SCADA system screens (Figure bellow)
My main concern for this article is the type of work presented here.
It does not look like a research article but more like a technical report on one particular system.
Response: We thank you for the comment and we understand your concern, but even though we do present a particular system, the presented work, if duplicated, could help others in building their own “one4all” framework.
Concerning the literature review. The positioning of one4all framework seems conceptually near to business intelligence. This type of systems could be considered to manage the sensor network?
Response: We thank you for the comment and the answer for your question it is that yes, it can be used to manage the sensor network, due to the fact that we already use the “one4all” framework to display the sensor network of the water company in the Maramures county. At the current time the displayed sensor network has not received any business intelligence, mostly because we didn’t have resources to go into that direction, but as you pointed (again, we thank for that) it could be something to do in the future.
Concerning the framework itself, is the code available somewhere? Due to the very technical point of view of this article, I would prefer to have access to the framework, to be able to reproduce the proposed platform.
Response: We thank you for the comment and we want to explain why we didn’t made the code available on the Internet. First of all, the code is a mix of database tables, SQL queries, Ignition dynamic templates and python code, being said that our framework it’s specific for Ignition SCADA systems. We can make all that mix available on github.com but most likely only a few companies that use Ignition and are in the water supplying business, would appreciate the code. Second, the purpose of our article was to present a concept that we were able to extract from building the “one4all” tool, using Ignition software, for the water supplying company in the county of Maramures. We thought that presenting a concept / an idea about how to build a “one4all” tool would help others in guiding them to do the same. We also tried in our article to prove the viability of the concept by presenting similar tools within the web-based programming vertical. We think that a tool, able to, first build and then modify at runtime an application, mostly by using description techniques rather than programming ones, will bring value to any company willing to work that way.

Reviewer 2 Report
This paper proposed a new SCADA approach. But there are some questions here:
- This paper presented the following purposes: "The main purpose of this paper is to introduce a new concept, named one4all” and "As a secondary purpose, the paper presents an integration of such a SCADA system with a GIS (Geographical Information System) system". But no novel concepts or innovations were provided except for some work description.
- "Due to the wide geographical operating range, a SCADA system needs to be put in place, but the management of such a system in a traditional way it’s hard to implement". Why the traditional management was hard to implement? What is the traditional management approaches? What are the advantages and disadvantages of the traditional appraoches? No such descripition was provided in this paper.
- In Section 3, some common methods were descripted. What is the main contributions of this work compared with the others?
- The example analyzed in this paper could not show the effectiveness of the approach. More experiments and results should be shared.
Author Response
We appreciate you for your precious time in reviewing our paper and providing valuable comments. Your valuable and insightful comments, led us to improve our paper. The authors have carefully considered the comments and tried our best to address every one of them. We hope that the manuscript, after careful revisions, will meet your high standards. The authors welcome further constructive comments if any.
Below we provide the point-by-point responses. All modifications in the manuscript have been highlighted in red.
Thank you again for your time and suggestions.
Response to Reviewer 1
Comments and Suggestions for Authors
This paper presented the following purposes: "The main purpose of this paper is to introduce a new concept, named one4all” and "As a secondary purpose, the paper presents an integration of such a SCADA system with a GIS (Geographical Information System) system". But no novel concepts or innovations were provided except for some work description.
Response: We thank you for the comment and will try to present our view. The new concept “one4all” it is the use of only one tool (elementary unit), able to generate views / screens, views combined into a SCADA system able to have different UI for every user, based on the geographic area where he or she is located. The new approach allowed us to build quickly and better. We thought that providing an example of our work will help others to understand the new proposed approach. We understand that our approach is a combination of old approaches, but we think that innovation also comes with combining old elements in a different format.
"Due to the wide geographical operating range, a SCADA system needs to be put in place, but the management of such a system in a traditional way it’s hard to implement". Why the traditional management was hard to implement? What is the traditional management approaches? What are the advantages and disadvantages of the traditional appraoches? No such descripition was provided in this paper.
Response: We thank you for the comment and will try to present our view. A traditional management involve creating screens for every station/SCADA element. Every screen (built from a template) needs changes, made by the SCADA developers (linking data tags, adding/deleting elements). Our approach is to have a tool (“one4all”) that will generate screens based on the high-level description made by users (not SCADA developers). That way, a company that has a SCADA system that is “alive”/changing and widely distributed, would not need to have many SCADA developers. Those companies will only need ordinary users with capabilities for adding new elements. We think the advantages are very clear, because finding users able to add SCADA elements it’s easier than finding SCADA developers. Furthermore, those users will be less paid than SCADA developers allowing companies to reduce costs.
In Section 3, some common methods were descripted. What is the main contributions of this work compared with the others?
Response: We thank you for the comment and will try to present our view. In Section 3 our intention was to present the elements used by us to build the SCADA “one4all” tool. We thought that providing those elements will help others to envision their own “one4all” tool.
The example analyzed in this paper could not show the effectiveness of the approach. More experiments and results should be shared.
Response: We thank you for the comment and will try to present our view. We created only 2 “one4all” tools, one for building a GIS application and another one for building the SCADA system. Each one is using a similar set of database tables for keeping the description of every screen.
We are very sorry but we don’t have other experiments.

Reviewer 3 Report
The paper proposes such a system, which integrates an existing SCADA system with the GIS and links the elements of the SCADA (plants, sensors) to their geographical position, by providing the user access with respect of one4all principle.
However, there are some technical details the authors may want to illustrate clearer to justify the quality of this paper.
- The author should illuminate clearly the methods the proposed system uses to prevent the data leakage, besides, please add more contents about the system security in the paper, which is very important in SCADA system.
- The authors fail to properly cite several past literatures highly related to this work (e.g., [1-3]) and carefully compare the differences between them and this paper.
[1] A Secure Revocable Fine-Grained Access Control and Data Sharing Scheme for SCADA in IIoT Systems, IOT Journal 2022.
[2] Secure Data Collection in Constrained Tree-Based Smart Grid Environments, SmartGridComm 2014.
[3] A Condition Monitoring and Fault Isolation System for Wind Turbine Based on SCADA Data, TII 2022.
- Please illuminate more clearly about your SCADA system scalability, and add some examples.
Author Response
We appreciate you for your precious time in reviewing our paper and providing valuable comments. Your valuable and insightful comments, led us to improve our paper. The authors have carefully considered the comments and tried our best to address every one of them. We hope that the manuscript, after careful revisions, will meet your high standards. The authors welcome further constructive comments if any.
Below we provide the point-by-point responses. All modifications in the manuscript have been highlighted in red.
Thank you again for your time and suggestions.
Sincerely,
Bogdan Vaduva
S.C. Vital S.A., SCADA/GIS department, Baia Mare, Romania
bogdan.vaduva@vitalmm.ro
Ionut Flaviu Pop
S.C. Vital S.A., SCADA/GIS department, Baia Mare, Romania
ionut.pop@vitalmm.ro
Prof. Dr. Eng. Honoriu Valean
Technical University of Cluj-Napoca, Automation Department, Cluj-Napoca, Romania
honoriu.valean@aut.utcluj.ro
Response to Reviewer 2
The author should illuminate clearly the methods the proposed system uses to prevent the data leakage, besides, please add more contents about the system security in the paper, which is very important in SCADA system.
Response: We thank you for the comment and we want to let you know that we used those citations because at the time we began our SCADA system we did some research which involved the use of SCADA systems (finding leakage) and the security of them. We did use those citations because those were the first ones found by us, of course each one at the appropriate time. For example, at the beginning of our SCADA endeavor, the entire system was in the same internal network, but in time the security concerns arose and we moved SCADA on a separate VLAN without internet and only a few entrance points. We are sorry if our good intentions (to provide the first sources of inspirations) were misunderstood.
We also made some changes within the article to let readers know about VLANs.
Please illuminate more clearly about your SCADA system scalability, and add some examples.
Response: We thank you for the comment and we added some explanations within the article.

Reviewer 4 Report
The paper presents a new concept and approach "one4all" in support of SCADA system, which is implemented through a software tool. The results and benefits of this software solution are well explained. Different generated visualizations regarding the users' role and permission is always in support of effective system control and management as well as in support of decision making.
I would suggest in the section Implementation a technical architecture of the software approach to be presented through a graphics.
Author Response
We appreciate you for your precious time in reviewing our paper and providing valuable comments. Your valuable and insightful comments, led us to improve our paper. The authors have carefully considered the comments and tried our best to address every one of them. We hope that the manuscript, after careful revisions, will meet your high standards. The authors welcome further constructive comments if any.
The paper presents a new concept and approach "one4all" in support of SCADA system, which is implemented through a software tool. The results and benefits of this software solution are well explained. Different generated visualizations regarding the users' role and permission is always in support of effective system control and management as well as in support of decision making.
I would suggest in the section Implementation a technical architecture of the software approach to be presented through a graphics.
Response: It was our pleasure to have you as our reviewer, due to your helpful and insightful comments. We’ll try to add to the manuscript the required information, so it can meet the standards required by any scientific paper.
The system architecture is as follows:
- One “one4all” tool/framework that incorporates all the dynamic templates used by the SCADA system
- One database server that contains the description and permission of generated SCADA system screens (Figure bellow/ attached file)
Below we provide the point-by-point responses. All modifications in the manuscript have been highlighted in red.
Thank you again for your time and suggestions.

Round 2
Reviewer 2 Report
- In Section 3, the concept should be showed using a figure.
- The added tables should be placed in implmentation section not in concept section. They are not concept.
Author Response
We appreciate your precious time in reviewing our paper and providing valuable comments. Your valuable and insightful comments, led us to improve our paper. The authors have carefully considered the comments and tried our best to address every one of them. We hope that the manuscript, after careful revisions, will meet your high standards. The authors welcome further constructive comments if any.
Below we provide the point-by-point responses. All modifications in the manuscript have been highlighted in red.
Thank you again for your time and suggestions.
Response to Reviewer
Comments and Suggestions for Authors
In Section 3, the concept should be showed using a figure.
The added tables should be placed in implementation section not in concept section. They are not concept.
Response: We appreciate your insightful suggestion and agree that it would be useful to move the Figure 1 and tables description into Implementation section. We thank you again for your comments and suggestions and you can see the changes into the revised manuscript.

This manuscript is a resubmission of an earlier submission. The following is a list of the peer review reports and author responses from that submission.
Round 1
Reviewer 1 Report
Thank you for giving me the opportunity to review this manuscript. The project and the solution proposed by the authors is very interesting, with high practical impact. the main issue of the manuscript is that the authors do not build it it as a scientific paper. That is why, in my opinion, it has to be restructured.
1- the abstract has to be rewrote - it should include the aim of the research, methodology, findings and the add value to the science
2- a subchapter with a literature review is needed. It has to be correlated with the practical aspects presented in part 2. I can agree that you identified a practical need, but you should demonstrate that there is a scientific/theoretical gap as well.
3- Section 3 - should start with the aim and hypothesis of the research ( a suggestion if I understand well - the hypothesis was to build a SCADA application with GIS capabilities, able to supervise simultaneous multiple, spread operational places). Your practical approach will consolidate the theoretical one.
4- Results - please describe the concept of the SCADA “one project to
run them all” (suggestion - you can name it SCADA one4all - is short and conceptualize) , followed by the practical description.
5 - Highlight the theoretic/scientific importance of your solution, generality, potential to be replicate by other water supply, gas, electricity suppliers or other companies that have to address the same need (for sure with specific sensors and adjustments to the particularities).
6 - The references must be improved
As a final remark - As it is the manuscript can be considered an acceptable practical project implementation - case study. The question is Why is of international interest of the academic environment what is happening in the water supply company in Maramures? Giving the theoretic/scientific dimension your idea it become an inspiration model for other developers.
Reviewer 2 Report
In this article, the authors introduce the SCADA project.
However, I will comment on some aspects of the manuscript:
-The manuscript does not contain enough to be considered in a journal.
-The titles of the Sections and Subsections must be improved.
-The Sections are not in the correct verb tense for a scientific article.
-Introduction Section is too short
-There is no Related Works Section
-It must improve the quality of the article.
-It does not have a correct scientific language.
-Figure 3 is an image or Tabala?
-Few results, there is no substantial contribution.
-The conclusions are small and must be improved. Also, it does not include Future Works.
-Improve References.
Round 2
Reviewer 1 Report
The authors considered the suggestions and significantly improved the paper. Also, I glad that I could contribute to the presentation of this SCDA concept.
Honesty, I wish the reference list to be larger and to clearly mention in the introduction the hypotheses, but they are not compulsory.
Reviewer 2 Report
Thanks to authors for performing changes suggested. However, the recommended length of an Article is more than 16 journal pages and the article presented do not cumplish with this requirement. Authors must review the Instructions for Authors (https://www.mdpi.com/journal/sensors/instructions).